# Snow-Disaster Risk Zoning and Assessment in Heilongjiang Province

**Hao Li, Wenshuang Xi, Lijuan Zhang * and Shuying Zang ***

Heilongjiang Province Key Laboratory of Geographical Environment Monitoring and Spatial Information Service in Cold Regions, Harbin Normal University, Harbin 150025, China; killerlight9023@163.com (H.L.); xws202112@163.com (W.X.)
* Correspondence: zlj@hrbnu.edu.cn (L.Z.); zsy6311@163.com (S.Z.)

**Abstract:** Heilongjiang Province is located in the northeast region of China, with the country's highest latitude. It has long and cold winters, and a temperate monsoon climate. Its unique geographic location and climatic conditions make it the second largest stable snow-covered region in China. The winter snow period starts in October and ends in April of the following year. Therefore, the long-term accumulation of snow causes road obstructions and low-temperature frost damage, which seriously affects local economic development and human safety. This study adopts snow parameters (e.g., snow depth and snow-cover period), natural environmental factors (e.g., elevation and slope), and socioeconomic factors (e.g., gross domestic product and light index). On the basis of the disaster risk assessment theory, we constructed a disaster risk index from four aspects (i.e., disaster risk, susceptibility, vulnerability, and disaster prevention and mitigation). Then, we performed snow-disaster risk zoning and an assessment in Heilongjiang Province. The main findings are as follows: the snow-disaster risk in the northern and eastern regions of Heilongjiang Province was high; the central and northern regions were highly sensitive to disasters; the main urban areas were highly vulnerable; and the economically developed regions had strong disaster prevention and mitigation capabilities. Overall, the spatial distribution of the snow-disaster risk followed a decreasing trend from east to west. High-risk areas were distributed in the east and northwest (covering 34.3% of the entire Heilongjiang Province area); medium-risk areas were distributed in the north and center (accounting for 45.2% of the entire Heilongjiang Province area); and low-risk areas were concentrated in the west (constituting 20.5% of the entire Heilongjiang Province area).

**Keywords:** snow disaster; risk assessment; risk zoning; Heilongjiang Province

## 1. Introduction

Since the 1990s, the focus of emergency management has gradually turned from emergency rescue and postevent recovery and reconstruction, to prepreventive preparations [1]. The United States, Britain, Germany, France, Japan, and other countries have promoted major disaster risk assessments. In 2004, the United Nations International Strategy for Disaster Reduction (UNISDR) and the United Nations Development Programme (UNDP) released two reports, respectively titled: "Living with Risk: A Global Review of Disaster Reduction Initiatives", and "Disaster risk reduction: a development concern". In March 2015, the "Sendai Framework for Disaster Risk Reduction 2015–2030", finally adopted by the Third World Conference on Disaster Risk Reduction, pointed out that two of the four priorities for disaster reduction are "Understanding disaster risk" and "Strengthening disaster risk governance to manage disaster risk" [2,3]. A snow disaster is large-scale snow, caused by heavy snowfall, which seriously affects the survival and health of humans and livestock. It is a meteorological disaster that can affect and damage traffic, communications, agriculture, and electricity [4]. In 1977, the snow disaster in Xilingol (Inner Mongolia) caused more than 70% of livestock deaths [5]. From 2000 to 2012, the agriculture sector of Liaoning Province suffered 20 large-scale disasters resulting from snowstorms [6]. In 2008,

the 100-year snow disaster swept over half of China, and affected circuits, communications, the water supply, and heating, to different degrees. According to statistics, the snow disaster caused 129 deaths; the emergency resettlement of 1.66 million people; 485,000 house collapses; and 178 million mu of crop damage; and resulted in direct economic losses of CNY 151.65 billion. The heavy snow in Heilongjiang Province in 2007 caused 754 houses to collapse. Local snowstorms affected more than 900 people in Suifenhe City and other places, with direct economic losses of nearly CNY 100 million [7]. It is evident that snow disasters have had a huge impact on human society. Therefore, snow-disaster risk assessment provides an important theoretical basis for predisaster preparations, as well as for scientifically setting the meteorological disaster prevention standards for various regional infrastructures, which is more conducive to promoting the construction of resilient cities and villages, and to fundamentally improving the ability to resist snow disasters.

Snow disasters occur in various forms, such as snow melting leading to avalanches and ice floods, less (more) snow leading to black (white) disasters, and abnormal snowfall in windy weather leading to snowstorms. The existing studies on snow-disaster risk mainly focus on three aspects: (1) Studies on single snow-disaster risks. Some scholars have evaluated the avalanche risk. Schmitt et al. assessed the avalanche risk on the Alps on the basis of topographical parameters, such as slope, aspect and elevation, forestry-related variables, and rocks [8]. Seliverstov et al. [9] carried out avalanche risk zoning in Russia on the basis of the recurrence interval of avalanches (avalanche frequency), the percentage of the whole investigated territory that is occupied by avalanche-prone areas, the duration of the avalanche danger period, the probability of a person's stay in an avalanche-prone area for 1 day (24 h) and for 1 year, and the total population of the area and its density. Cappabianca et al. [10] presented an avalanche risk estimation procedure that combines a statistical analysis of the snowfall record, iterative simulations of avalanche dynamics, and empirically based vulnerability relations. Germain et al. [11] reconstructed past avalanche events in the north of the Gaspe Peninsula on the basis of tree rings and assessed the avalanche risk. In 2016, Germain also analyzed the avalanche risk from natural factors, the population, and environmental factors for avalanche disaster in northern Quebec, eastern Canada [12]. Some scholars have studied the risk of snowstorm disasters. Zhang et al. [13] calculated the probability of each level of the snowstorm information diffusion theory on the basis of the daily snow precipitation in 63 cities and counties in Heilongjiang Province, China, from 1961 to 2015, and established a hazard index model by using the snowstorm probability and the amount of snow precipitation. Next, a hazard assessment and the regionalization of snowstorms was performed for Heilongjiang Province from 1961 to 2015, and it was proposed that the high-risk areas increased by 30.7% from the 1960s to the 2010s, as opposed to the 38.9% reduction in the low-risk areas. Liu et al. [14] constructed the snowstorm disaster risk index on the basis of the frequency of snowstorms and assessed and regionalized the snowstorm risk in northeast China in the future (2020–2099). They showed that, under the RCP4.5 and RCP8.5 scenarios, the areas of the high-risk and light-risk regions would increase, while the areas of the low- and medium-risk regions would decrease. (2) Studies on the impact of snow disasters. Snow disasters have a major impact on agriculture, animal husbandry, and road transportation. Barbolini et al. [15], on the basis of avalanche accidents that occurred during outdoor winter activities over the Italian Alps in the last 20 years, proposed a vulnerability relation for people directly exposed to avalanches. Sinickas et al. [16] discussed the effects of the long-term changes in the avalanche occurrence rates in terms of consequences and vulnerability. Casteller et al. [17] report on the relationship between Nothofagus broadleaf forests and avalanche runout distances. Gao [18] constructed an agricultural risk estimation index and divided the risks of China's ice and snow on agriculture. Their results propose that the regions with high snow and freezing occurrences are located in northwestern China, and that the regions with the high-loss areas are located in the coastal and southeastern parts. Sa et al. [19] established a snow-cover index using passive microwave remote sensing data from 1978 to 2012, and evaluated the risk factors of disaster in the pastoral areas of the Inner Mongolia

Grasslands. Liu et al. [20] constructed the snow-disaster index and the vulnerability index by using the highway network, social and economic development data, meteorology data, and ArcGIS software of Guoluo Prefecture, Qinghai Province, combined with the analytic hierarchy process (AHP) and cluster analysis, to calculate the snow-road-disaster risk index of Guoluo Prefecture. Qi et al. [21] proposed the risk assessment index system of highway snow cover on the basis of a theoretical model and put forward a distribution of trunk highway snow-disaster risks in Shaanxi. Leone et al. [22] evaluated the impact of avalanches on road traffic in three Alpine departments (Alpes-de-Haute-Provence, Hautes-Alpes, and Alpes-Maritimes). Snow disasters can also amplify the effects of other disasters, such as earthquakes [23]. (3) Studies on the comprehensive risk regionalization of snow disasters. Wang et al. [24] further analyzed the formation mechanism of snow disasters (SDs), and constructed the integrated risk index (IRI) of a snow disaster from the aspects of historical disasters, snowfall events, disaster-formation environments, livestock overload, and livestock vulnerability and adaptability. The regions with high IRIs are mainly concentrated in the middle, east, and southwest of the Qinghai-Tibet Plateau, and appear as a contiguous risk belt from northeast to southwest. Liu et al. [25] conducted a comprehensive analysis of the 18 indexes of snow disaster on the Qinghai-Tibet Plateau, based on the hazard harmfulness data collected from historical records and the data collected from the entities affected by this hazard in 2010, and classified the snow-disaster hazards. Gao et al. [26], on the basis of the logistic regression of 33 snow-disaster events in Qinghai province, with the maximum snow depths, snow-cover days (SCDs), slopes, annual average temperatures, and per capita gross domestic product (GDP), constructed a potential risk assessment model to regionalize the snow-disaster risk in Qinghai Province. Park et al. [27] selected the pressure index (PI), the state index (SI), and the response index (RI) to assess the comprehensive risk of snow disaster in the metropolitan city of Ulsan. Insang [28] also assessed and analyzed the snow disaster risks of Daegu City, Ulsan City, Gyeongsangbuk Province, and Gangwon Province on the basis of the subindicators of three hazards, six exposures, four vulnerabilities, and five adaptive capacities.

In summary, scholars have conducted a lot of studies on snow-disaster risk and its impact. In addition to studies on the risk of a single snow disaster, studies on snow-disaster risk regionalization, from the perspectives of risk, sensitivity, vulnerability, and disaster prevention and mitigation, have gradually increased in recent years. However, the existing risk studies of snow disaster are mostly based on snow-cover data, and the risk indicators of snowstorms and wind-blown snow are seldom considered. Therefore, the risk of snow disaster in this paper comprehensively considers snow, snowstorms, and wind-blown snow. Compared with previous studies, the index selection has certain characteristics and is comprehensive. Heilongjiang Province is the highest-latitude area in China, belonging to the second largest snow-covered area in China. It is different from other snow-covered areas because of its large annual average snow reserves and an obvious interannual variability [29]. Studies have shown the increasing intensity of winter snowfall in Heilongjiang Province in recent years [30]. The years with abnormal snowfall have also increased, especially in the first decade of the 21st century. Extreme winter precipitation events in Heilongjiang Province have frequently occurred, and the precipitation values in 2002, 2003, 2009, and 2010 were over 50% more than those in normal years [31]. Heilongjiang Province is the region with the highest average annual wind speed in China [32]. The secondary disasters caused by wind and snow are greater [7], and the persistence of snow disasters is more significant than in other regions. Therefore, we selected Heilongjiang Province as the study area. According to the natural disaster risk theory model, we established a snow-disaster risk index according to the disaster risk, susceptibility, vulnerability, and the disaster prevention and mitigation capability. Combined with geographic information system (GIS) spatial analysis tools, the weighted comprehensive evaluation method, and the analytic hierarchy process, this study classifies the risk zoning of snow disasters in Heilongjiang Province. It provides a quantitative reference for Heilongjiang province from which to determine the snow-disaster prevention

standard, make a disaster prevention plan, and assess the losses after a disaster. It is also useful for carrying out snow-disaster risk assessments and regionalization-related research.

## 2. Materials and Methodology

### 2.1. Study Area

Heilongjiang Province, located in northeastern China (121°11′–135°05′ E, 43°26′–53°33′ N), is the most northernly territory of the country (Figure 1). It borders Russia in the north, along Heilongjiang River. Characterized by a temperate continental monsoon climate, it has long cold winters and short summers. The annual average temperature ranges from −4 °C to 5 °C, from north to south of the study area. Its total annual precipitation is around 400–650 mm, which gradually decreases from east to west. The average annual wind speeds are about 2–4 m/s. Heilongjiang Province is an important grain production base in China, with commercial grain accounting for about one-tenth of the national total.

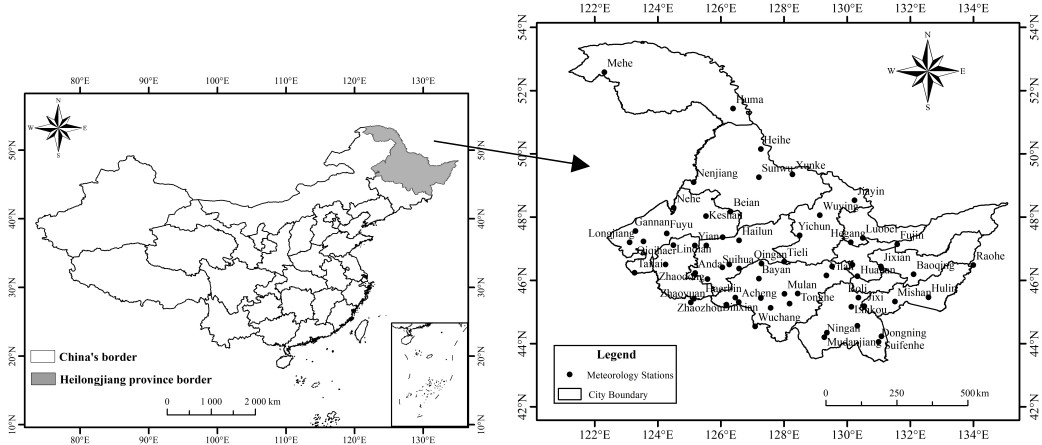

**Figure 1.** The study area and distribution of 62 meteorological stations in Heilongjiang Province, China.

### 2.2. Datasets

Precipitation and wind speed data: The daily meteorological data used in this study were provided by the Heilongjiang Meteorological Data Center. The earliest meteorological station in Heilongjiang Province was built in 1951, and 83 stations were established later. They have been divided into national standard meteorological stations and general meteorological stations. According to the continuity of data and the number of stations required for analysis, we selected 62 stations with continuous and complete observation data since 1961. The spatial distribution of the stations is shown in Figure 1. The basic data used in the study include the daily temperatures and precipitation data. The time scale is from 1983 to 2020. The study period is winter, which is defined as taking place from November of the current year to February of the second year.

Elevation, slope, and other data: The elevation, slope, and lighting data were generated by visual interpretation from the Resource and Environment Science and Data Center (www.resdc.cn. Accessed on 9 February 2021), with a spatial resolution of 90 m. The terrain standard deviation was calculated on the basis of the elevation.

Normalized difference vegetation index (NDVI) data: The NDVI data were downloaded from the National Aeronautics and Space Administration (NASA) website, from November of the current year to February of the following year, with a resolution of 1 km × 1 km. The time period is from 1996 to 2020, which is the winter average of the past five years.

Nighttime light index data: This study used national stable light images. Briefly, a stable light image is an annual raster image that calibrates the average nighttime light intensity. Each image included the lights of cities, towns, and the permanent light sources of other places (excluding the influence of occasional noises, such as moonlight clouds, firelight, and oil and gas burning). The pixel DN value of the image represents the average light intensity, and its range was 0–63. The nighttime light index data were downloaded from the Geographical Information Monitoring Cloud Platform (http://www.dsac.cn/DataProduct/Detail/201116. Accessed on 13 January 2021).

Socioeconomic factors: The socioeconomic factors (e.g., gross domestic product [GDP], per capita disposable income, and education level) used in this study were from the Heilongjiang Statistical Yearbook 1996–2020 (Accessed on 10 January 2021).

*2.3. Methods*

On the basis of the basic theory of disaster risk assessment, this paper constructs a theoretical model of a snow-disaster risk assessment. The trend analysis method is used to analyze the time characteristics of the risk factors; the analytic hierarchy process is used to determine the weights of the risk-influencing factors, and the comprehensive weighted evaluation method is used to form a risk index model.

2.3.1. Basic Theory of Disaster Risk Assessment

According to the theory of natural disaster risk formation, meteorological disaster risk is formed by four parts: risk, susceptibility (hazard-bearing object), vulnerability (hazard-bearing object), and the disaster prevention and mitigation capability. Each factor is composed of a series of subfactors. Its expression is:

$$\text{Disaster risk index} = f \text{ (risk, susceptibility, vulnerability, disaster prevention and mitigation capability)} \tag{1}$$

The risk factors: All meteorological factors that may cause disasters can be called "meteorological hazards", and most of the meteorological hazards that exist in disaster-causing environments are abnormalities of certain natural phenomena, temporal and spatial laws, or natural substances, or a certain abnormality in the process of energy exchange. Generally, the greater the risk of meteorological factors, the greater the risk of meteorological disasters.

Susceptibility of the disaster-bearing object: The disaster-bearing object is the target of the hazard-causing factor and is the entity that bears the disaster. The susceptibility of the hazard-bearing object is the property shown by the hazard-bearing individual exposed to the hazard-causing agent and is the result of the interaction between the hazard-causing factor and the hazard-bearing object.

Vulnerability of disaster-bearing object: Disasters can only become disasters when they act on corresponding objects, namely, humans and their social and economic activities. Specifically, a disaster-bearing object refers to all objects that may be threatened by hazards in a given dangerous area, as well as to the degree of damage or loss caused by potential hazards, which comprehensively reflects the degree of loss of meteorological disasters.

The disaster prevention and mitigation capability refers to the various management measures and countermeasures that are used to prevent and mitigate meteorological disasters, including management capabilities, disaster-reduction investment, resource preparation, etc. With the proper management measures and strong management capabilities, the smaller the potential loss that may be suffered, and the smaller the risk of meteorological disasters.

On the basis of the above theory, we built a hierarchical analysis model for snow-disaster risk assessment (Figure 2). Please refer to Sections 3.1–3.4 for the selection basis of the specific indicators.

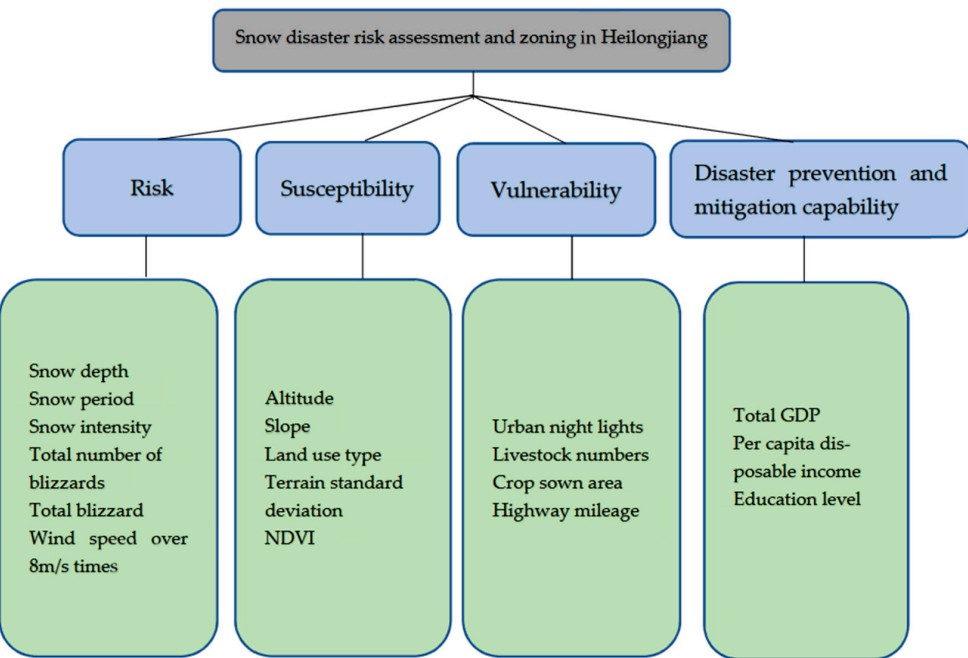

**Figure 2.** Hierarchical model of snow-disaster risk assessment.

### 2.3.2. Weighted Comprehensive Evaluation Method

The weighted comprehensive evaluation method is a method for solving the "bottom-up" indexes in the risk hierarchy analysis and the evaluation model [33]. This method comprehensively considers the degree of influence of each factor on the overall object and integrates the advantages and disadvantages of each specific index, using a numerical index to concentrate the data, indicating the pros and cons of the entire evaluation object. Therefore, this method is particularly suitable for the comprehensive analysis, evaluation, and optimization of technologies, strategies, or programs, and is currently one of the most commonly used calculation methods. Its expression is:

$$Y_i = \sum_{j=1}^{m} \lambda_{ij} X_{ij} \quad i = 1, 2, 3, 4; \quad j = 1, 2, \ldots, m \tag{2}$$

In the formula, $Y_i$ represents the disaster risk index, and $i$ represents, respesctively, the risk, susceptibility, vulnerability, and disaster prevention and mitigation capabilities; $X_{ij}$ is the factor that affects the risk, susceptibility, vulnerability, and disaster prevention and mitigation capability; and $\lambda_{ij}$ is the weight value of the risk, susceptibility, vulnerability, and disaster prevention and mitigation capability ($0 \leq \lambda_j \leq 1$).

The expression for the comprehensive risk index of natural disasters is:

$$Y = \sum_{i=1}^{n} W_i Y_i \quad i = 1, 2, 3, 4 \tag{3}$$

In the formula, $Y$ represents the comprehensive disaster risk index; $Y_i$ is the risk index, the susceptibility index, the vulnerability index, and the disaster prevention and mitigation capability index; and $W_i$ is the weight value. The stronger the disaster prevention and mitigation ability, the smaller the comprehensive risk index and, thus, the "negative sign" is used.

The $\lambda_j$ and $W_i$ are determined by the analytic hierarchy process (see research method in Section 2.3.3 for details); each factor in the formula needs to be standardized because of different dimensions (see research method in Section 2.3.4 for details).

### 2.3.3. Analytic Hierarchy Process

The analytic hierarchy process (AHP) is a simple method for making decisions on some more complex and vague problems, especially for those problems that are difficult to fully quantitatively analyze [34]. This paper uses the operation principle of the analytic hierarchy process and uses the 1–9 scale method, provided by Saaty, to construct the judgment matrix for the pairwise relationships of the influence factors. The pairwise comparison of all of the influence factors determines the weight of each influence factor, which avoids the result error caused by the subjectivity of the expert. The qualitative comparison scale values between the two influencing factors are shown in Table 1 below.

**Table 1.** Scale of the AHP method.

| Scale $b_{ij}$ | Definition |
| :---: | :---: |
| 1 | The *i* factor is as important as the *j* factor. |
| 3 | The *i* factor is slightly more important than the *j* factor. |
| 5 | The *i* factor is more important than the *j* factor. |
| 7 | The *i* factor is much more important than the *j* factor. |
| 9 | The *i* factor is absolutely more important than the *j* factor. |
| 2, 4, 6, 8 | Between the noted levels. |

Solve the maximum eigenvector value of the judgment matrix and its corresponding eigenvector by the sum-product method and check the consistency of the matrix (the following formula). After passing, solve it by the sum-product method.

$$CI = \frac{\lambda_{max} - n}{n - 1} = \frac{-\sum\limits_{i=1}^{n} \lambda_i}{n - 1} \tag{4}$$

$$CR = \frac{CI}{RI} < 0.1 \tag{5}$$

In the formula, *CI* is the consistency index of the judgment matrix; $\lambda_{max}$ is the largest characteristic root of the matrix; *n* is the order of the discrimination matrix; *CR* is the random consistency index of the judgment matrix; *RI* is the average random consistency index of the discrimination matrix. The *RIs* of the risk, susceptibility, vulnerability and the disaster prevention and mitigation capability are 1.24, 1.12, 0.9, and 0.58, respectively.

According to the analytic hierarchy process, taking risk as an example, the process of determining the weight of each factor is as follows:

The hazard factors include the snowfall intensity, snow depth, snow-period length, snowfall days, blizzard volume, blizzard number, and the days with wind speeds over 8 m/s. It is believed that, according to the degree of importance, the relative hazard factors of the blizzard amount are the most important, followed by the number of blizzards, the snowfall intensity, the snow depth, the snow-cover period, the snowfall days, and the wind speeds. According to the relative importance of each factor, the judgment matrix is constructed according to Table 1. The maximum characteristic root of the matrix, $\lambda_{max}$, is 6.242, *CR* = 0.038 < 0.10, and the weight result is valid. Therefore, the weights of the snow intensity, snow depth, snow period, the total blizzard volume, the total number of blizzards, and wind speeds over 8 m/s are 0.3848, 0.3054, 0.1648, 0.0718, 0.0468, and 0.0264, respectively.

In the same way, the weights of the susceptibility, vulnerability, disaster prevention and mitigation capability, and the comprehensive disaster risk factors are shown in Table 2.

**Table 2.** The weights of the risk, susceptibility, prevention and mitigation capacity, and comprehensive risk.

| Sensitivity | Elevation 0.4635 | Slope 0.282 | NDVI 0.1448 | TSD 0.0727 | Land Use Type 0.037 |
|---|---|---|---|---|---|
| Susceptibility | UNL 0.4742 | Highway mileage 0.303 | livestock numbers 0.154 | crop sown area 0.0689 | |
| Prevention and mitigation capacity | Total GDP 0.4885 | Education level 0.1994 | Per capita disposable income 0.3121 | | |
| Comprehensive risk | Risk 0.4627 | Susceptibility 0.3272 | Vulnerability 0.1357 | Prevention and mitigation capacity 0.0744 | |

Note: TSD (terrain standard deviation); UNL (urban night lighting).

### 2.3.4. Standardization

In the process of zoning, because of the different dimensions of the selected factors, the magnitude of the difference is large. For example, the length of the snow-cover period is 150 days, and the average number of snowfalls is about 10 times per year. Therefore, when calculating the hazard-factor risk index, it needs to be normalized so that the value of each factor is between 0 and 1. When assessing the hazard risk, the hazard-bearing body susceptibility, the hazard-bearing body vulnerability, the disaster prevention and mitigation capability, the relationships between the selected influencing factors and the hazard-causing factor risk, the hazard-bearing body susceptibility, the hazard-bearing body vulnerability, and the disaster prevention and mitigation capability are different. Some are the greater the number of influencing factors, the greater the hazard risk, the greater the susceptibility of the hazard-bearing body, the greater the vulnerability of the hazard-bearing body, and the greater the ability of disaster prevention and mitigation, while some factors are the opposite. Therefore, in the evaluation process, the maximum value standard or the minimum value standardization are performed for ostentation, and an example of the formula is as follows: The greater the snowfall intensity, the greater the hazard risk of the hazard factor. Therefore, the maximum value of the snowfall intensity is standardized, and Formula (6) is selected; if the highway mileage is longer, the vulnerability is smaller and, thus, the highway mileage is minimized. For the value standardization, select Formula (7).

Maximum standardization:

$$X'_{max} = \frac{|X_{ij} - X_{min}|}{X_{max} - X_{min}} \tag{6}$$

Minimum standardization:

$$X'_{min} = \frac{|X_{max} - X_{ij}|}{X_{max} - X_{min}} \tag{7}$$

where $X_{ij}$ is the index number of the *j*-th factor of the *x* factor; $X'_{max}$ and $X'_{min}$ are the dimensionalities of $X_{ij}$; and $X_{max}$ and $X_{min}$ are the minimum and maximum values in the index sequence, respectively.

### 2.3.5. Mann–Kendall Trend Test

In order to test the change trend of the meteorological elements, the Mann–Kendall trend test method is used to test the meteorological elements on the annual and seasonal scales [35]. The method assumes that the time data series $(x_1, x_2, \ldots, x_n)$ are independent, random, and uniformly distributed. For the sample sequence, the calculation equation for the test statistical variable, *S*, is:

$$S = \sum_{i=2}^{n} \sum_{j=1}^{i-1} sign(x_i - x_j) \tag{8}$$

when:

$$(x_i - x_j) = \left\{ \begin{array}{c} > 0 \\ = 0 \\ < 0 \end{array} \right\}, sign(x_i - x_j) = \left\{ \begin{array}{c} 1 \\ 0 \\ -1 \end{array} \right\} \tag{9}$$

$$\sigma s = \sqrt{\frac{n(n-1)(2n+5)}{18}} \tag{10}$$

when:

$$S \left\{ \begin{array}{c} > 0 \\ = 0 \\ < 0 \end{array} \right\}, Z = \left\{ \begin{array}{c} \frac{S-1}{\sigma s} \\ 0 \\ \frac{S+1}{\sigma s} \end{array} \right\} \tag{11}$$

where $S$ is the test statistical variable of the normal distribution; $x_i$ and $x_j$ are two series of different distributions in the same sample, where $1 \leq j < I \leq n$; $\sigma_S$ is the standard deviation; $n$ is the total number of samples; and $Z$ is the test value. If $Z > 0$, the tested time series has an upward trend, and if $Z < 0$, the tested time series has an upward and downward trend; the absolute value of $Z$ is greater than or equal to 2.32, 1.64, and 1.28, which means they pass confidence. They are, respectively, 99%, 95%, and 90% significance test levels.

### 2.3.6. The Pettitt Test

The Pettitt test is a common tool to detect a single unknown mutation point, and it is also one of the most common nonparametric test methods [36]. It can be described as giving an observation data sequence, $X \mid x_t$, $t = 1, 2, \ldots, n$, where n is the sample size, if the sequence has a change point at $\tau(1 \leq \tau \leq n - 1)$, defining the corresponding Pettitt statistics, $U_\tau$.

$$U\tau, n = \sum_{i=1}^{\tau} \sum_{j=\tau+1}^{n} sgn(x_i - x_j) = \left\{ \begin{array}{l} -1(x_i - x_j) < 0 \\ 0(x_i - x_j) = 0 \\ +1(x_i - x_j) > 0 \end{array} \right. \tag{12}$$

Let $K = \max(|U_{\tau,n}|)$, then the time, $T$, where $K$ is located, is the possible change point position. The significance level of the corresponding change point is:

$$P = 2exp\left[-6K^2(n^3 + n^2)\right] \tag{13}$$

If $P \leq 0.5$, then time, $T$, is considered to be the position of the statistically significant change point.

## 3. Results

### 3.1. Snow-Disaster Risk Analysis

3.1.1. The Characteristics of the Risk Zoning Factors of Snow Disasters in Heilongjiang Province

In this paper, the snow depth, the length of the snow-cover period, the snowfall intensity, the total blizzard volume, the total number of blizzards, and the total days of wind speeds over 8 m/s were selected as the risk factors of snow disaster. The deeper the snow depth, the longer the snow-cover period, the stronger the snowfall intensity, the greater the blizzards, the greater the total number of blizzards, and the more days with wind speeds over 8 m/s, the greater the risk of snow disasters. Figure 3 shows the interannual variation characteristics of the snow depth, the snow-period length, the snow intensity, the total blizzard volume, the total number of blizzards, and the days with wind speeds over 8 m/s in Heilongjiang Province, from 1983 to 2020. It can be seen that the snow depth, snow intensity, total blizzard volume, and the total number of blizzards in Heilongjiang Province show a significant increase, and the M–K statistics are 2.869, 3.269, 1.968, 2.014, respectively. The length of the snow-cover period and the wind speeds showed a significant decrease, and the M–K statistical values were −3.223 and −6.43 (Table 3). It can be seen from the distribution of the box map (Figure 4) that the snow depth, the snow-period length, the snow intensity, the total blizzard volume, the total number of blizzards,

and the days with average wind speeds over 8 m/s are 20.82 cm, 165.4 d, 1.24 cm/time, 142.0 mm, 9.9 times, and 92 d, respectively. The interannual changes in the wind speed, snow intensity, the number of blizzards, and the snow depth are relatively large, while the interannual changes in the snow depth and the blizzard volume are relatively small. It can be seen from the extreme value distribution characteristics that the maximum value of the snow intensity, the maximum value of the blizzard volume, and the number of blizzards are obviously larger than the average values, while the minimum value of the days with wind speeds over 8 m/s is obviously smaller than the average value. Using the Pettitt method to calculate the mutation year of each hazard factor, it was concluded that the snow depth, the snow-cover length, the snow intensity, and the days with wind speeds over 8 m/s days in Heilongjiang Province all have abrupt changes around the early 2000 s, and the catastrophes of the total snow cover and the total number of snowstorms occur in the middle and late parts of the 2000 s, respectively.

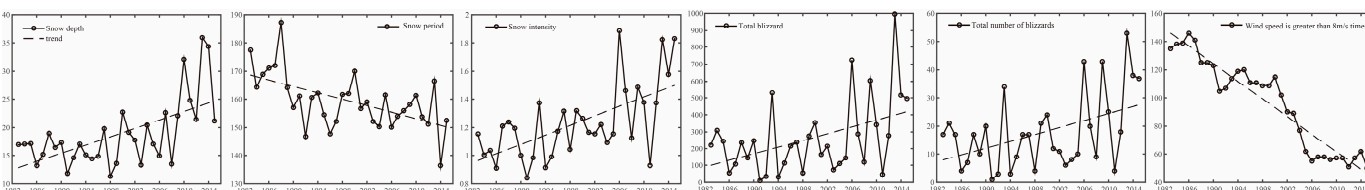

**Figure 3.** Time variation of risk factors.

**Table 3.** The M–K statistics and abrupt years of risk factors.

| Disaster-Causing Factors | Snow Depth | Snow Period | Snow Intensity | Total Blizzard Volume | Total Number of Blizzards | Wind Speed Is Greater than 8 m/s Times |
|---|---|---|---|---|---|---|
| M–K statistics | 2.867 ** | −3.223 ** | 3.269 ** | 1.968 * | 2.014 * | −6.43 ** |
| Abrupt year | 1999 | 2002 | 1998 | 2005 | 2008 | 2000 |

Note: "*", "**": significance at 0.05 and 0.01 levels, respectively.

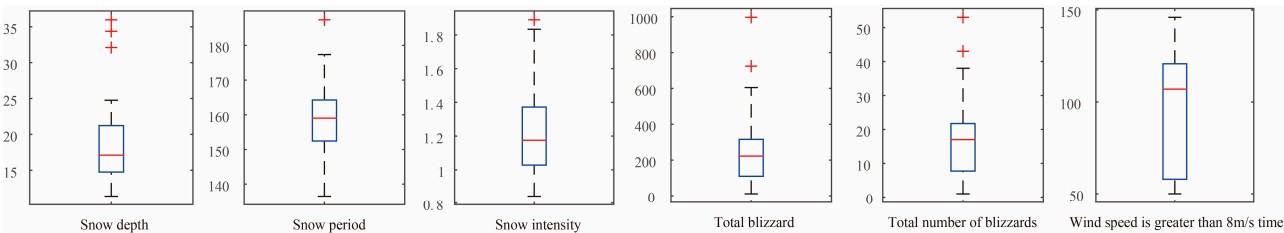

**Figure 4.** Boxplots of risk factors.

Figure 5 shows the spatial distribution of the snow depth, the snow-period length, the snow intensity, the total blizzard volume, the total number of blizzards, and the days with wind speeds over 8 m/s in Heilongjiang Province, from 1983 to 2020. It can be seen that the snow depths of Heilongjiang Province, from 1983 to 2015, are between 7.39 and 35.26 cm, and that the snow depths of 25–35.26 cm are mainly distributed in Mohe County in the northwest, northwest of Tahe County, northwest of Huzhong District, Xunke County, Jiayin County, north of Yichun City, north of Hegang City, and north of Luobei County; snow depths of 15–25 cm are mainly distributed in most cities and counties in central and southern Heilongjiang Province; snow depths of 7–15 cm are mainly distributed in the southwest of Heilongjiang Province, including Qiqihar cities and counties, Daqing, and the southwest of Suihua. The length of the snow period in Heilongjiang Province from 1983 to 2015 is 150–200.3 days. Snow periods of more than 180 days are mainly distributed in the Daxing'anling area, in the north of Heilongjiang Province; the areas of snow periods of more than 160 days are mainly distributed in central and northern Heilongjiang Province; the areas with a snow-cover period of less than 160 days are mainly

distributed in the southern part of Heilongjiang Province. The length of the snow-cover period of the province generally increases from low latitudes to high latitudes. The snow intensity in Heilongjiang Province from 1983 to 2020 was from 1.23–2.10 cm/time. The areas with higher snow intensities are mainly located in the east of Hegang, Jiamusi, and in Shuangyashan in the east; the areas with moderate snow are mainly located in the east of Heilongjiang Province and Mohe County; and the areas with low snow intensity are mainly distributed in the west of Heilongjiang Province. The total blizzards in Heilongjiang Province from 1983 to 2020 were between 0 and 761.9 mm, and the blizzards above 200 mm were mainly distributed in Jiamusi, Hegang, Shuangyashan, Qitaihe, Jixi, and Mudanjiang, in eastern Heilongjiang Province; the areas with blizzards above 100 mm were mainly distributed in the Yichun, eastern Suihua, eastern Harbin, Mudanjiang, and Daxing'anling regions, in the central part of Heilongjiang Province; the areas with blizzards below 100 mm were mainly distributed in Heihe, Qiqihar, Daqing, Suihua, and Harbin, in the western part of Heilongjiang Province. The total number of blizzards in Heilongjiang Province from 1983 to 2020 was between 5 and 20 times. The total number of blizzards above 15 were mainly located in Jiamusi, Hegang, Shuangyashan, Qitaihe, Jixi, and Mudanjiang, in eastern Heilongjiang Province; the number of blizzards were 10 in the areas mainly distributed in Qiqihar, Daqing, Suihua, and Harbin, in the west of Heilongjiang Province. In Heilongjiang Province, from 1983 to 2020, the number of days with wind speeds over 8 m/s were 70–187.7 days, and the days with wind speeds over 8 m/s over 110 days were mainly distributed in Jiamusi, Hegang, Shuangyashan, Qitaihe, Jixi, and Mudanjiang, in eastern Heilongjiang Province. The areas with wind speeds> 8 m/s within 90–130 days are mainly distributed in the Qiqihar, Daqing, Heihe, and Daxing'anling areas, in western Heilongjiang Province; the areas with wind speeds over 8 m/s below 90 days are mainly distributed in Heihe, Suihua, and Iraq, in central Heilongjiang Province.

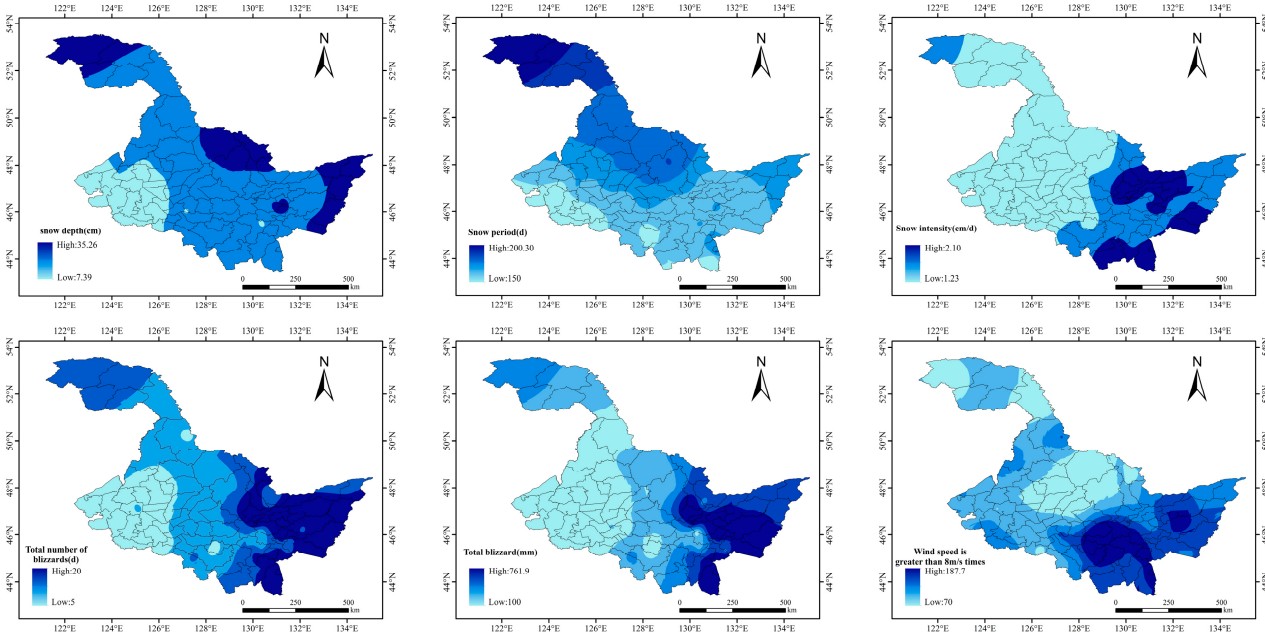

**Figure 5.** Distribution map of risk factors of snow disaster.

### 3.1.2. Results of the Risk Zoning of Snow Disasters in Heilongjiang Province

The spatial superpositions of the risk factors are used to obtain the disaster risk zoning results of snow disasters in Heilongjiang Province (Figure 6a). The spatial distribution of hazards is uneven, and there are high-value areas locally. The risk index is equally divided into three levels (as shown in Table 4), and the spatial distribution of the risk classification is presented in Figure 6b. The spatial distribution of the snow-disaster risk

in Heilongjiang Province followed a decreasing trend from east to west. High-risk areas accounted for 46.3% of the entire Heilongjiang area, and were distributed mainly in the eastern region of Heilongjiang (Jiamusi, Yichun, Jixi, Qitaihe, Hegang, Shuangyashan, Xunke, and Mudanjiang) and the northwestern Daxing'anling area; the medium-risk areas covered 36.8% of the entire Heilongjiang area, and were distributed primarily in the central region of Heilongjiang Province (eastern Daxing'anling, Heihe, eastern Suihua, northern Mudanjiang, and eastern Harbin); and the low-risk areas accounted for 16.9% of the total area of Heilongjiang Province, and were distributed mostly in the western region of Heilongjiang Province (Qiqihar, Daqing, western Suihua, and western Harbin).

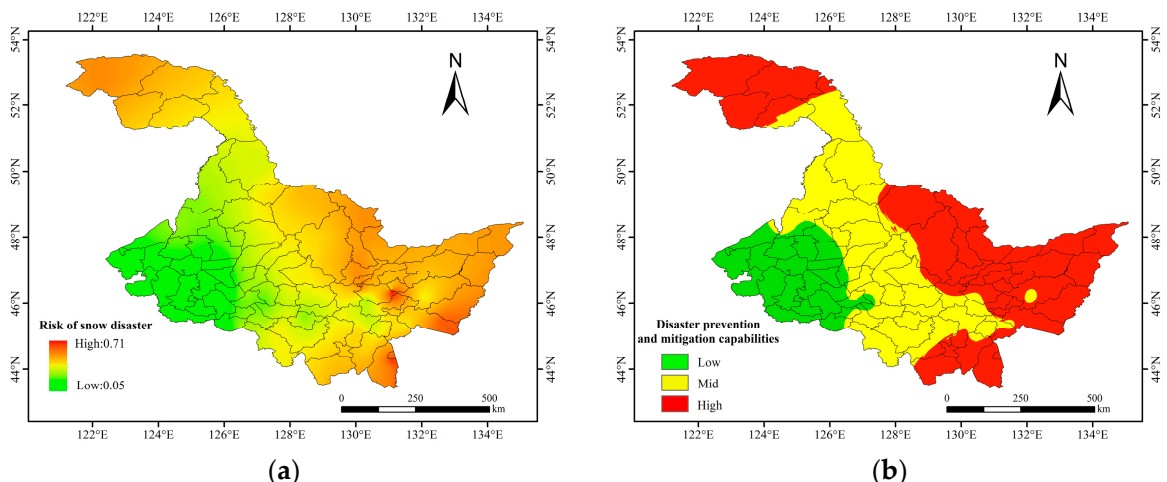

**Figure 6.** Risk map of snow disaster in Heilongjiang Province ((**a**): stretch, (**b**): classification).

**Table 4.** Risk classification criteria.

| Index | Low | Mid | High |
|---|---|---|---|
| Risk | 0.046–0.244 | 0.244–0.414 | 0.414–0.712 |
| Susceptibility | 0.014–0.129 | 0.129–0.245 | 0.245–0.667 |
| Vulnerability | 0–0.178 | 0.178–0.368 | 0.368–0.824 |
| Prevention and mitigation capacity | 0–0.178 | 0.178–0.280 | 0.280–0.969 |
| Comprehensive Snow Disaster Risk | 0.067–0.295 | 0.295–0.434 | 0.434–0.666 |

*3.2. Susceptibility Analysis of Snow Disaster*

3.2.1. The Characteristics of the Susceptibility Zoning Factors of Snow Disasters in Heilongjiang Province

Figure 7 shows the spatial distribution of the altitude, slope, land-use type, terrain standard deviation, and the NDVI in Heilongjiang Province. It can be seen that the altitude in Heilongjiang province is between 15 and 1651 m; the slope is between 0 and 57.4°; the terrain standard deviation is between 0 and 174.14; and the spatial distribution of the altitude, the slope, and the terrain standard deviation are roughly the same. The areas with high levels are mainly distributed in Huzhong District, Xinlin District, Yichun City, Hailin City, Muling City, and Dongning County in Heilongjiang Province. The land-use types in Heilongjiang Province are dominated by farmland and forest. The forest land is mainly distributed in the Greater Khingan Mountains, in the Lesser Khingan Mountains, and in Mudanjiang City in the south. The NDVI values corresponding to the forest area are also higher, and the highest is 0.83. Farmland is mainly distributed on the Songnen Plain, in the west of Heilongjiang Province, and on the Sanjiang Plain in the east.

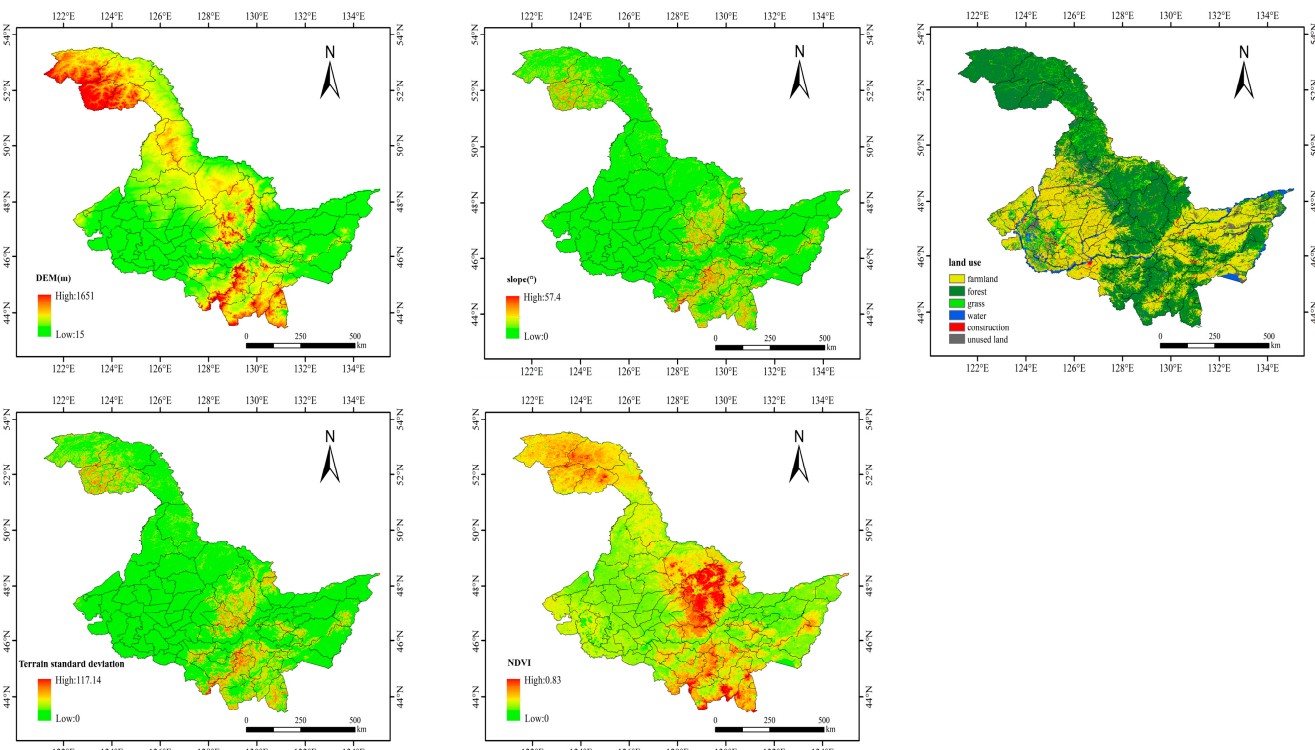

**Figure 7.** Heilongjiang Province altitude, slope, land-use type, terrain standard deviation, NDVI spatial distribution maps.

### 3.2.2. Results of the Susceptibility Zoning of Snow Disasters in Heilongjiang Province

The weight in Table 2 is used to perform the spatial superposition of the susceptibility factors to obtain the susceptibility zoning results of snow disasters in Heilongjiang Province (Figure 8a). The spatial distribution of the snow-disaster susceptibilities in Heilongjiang Province is high in the central and northwestern regions, and is low in the eastern and western regions. The susceptibility index is equally divided into three levels (as shown in Table 4), and the spatial distribution of the susceptibility classification is presented in Figure 8b. High susceptibilities accounted for 16.3% of the entire Heilongjiang area, and were distributed mainly in the northern region (southern Mohe County, Huzhong District, Xinlin District, Yichun City, Tieli City, and Mudanjiang City); medium susceptibilities covered 35.6% of the entire Heilongjiang area, and were distributed largely in the central region of Heilongjiang Province (Heihe City, Huma County, Tahe County, Jixi City, and Yichun); and low susceptibilities accounted for 48.1% of the total area of Heilongjiang Province, and were distributed mainly in the western (Qiqihar, Daqing, Suihua, and Harbin) and eastern (Jiamusi, Shuangyashan, and Jixi) regions of Heilongjiang Province. Heilongjiang Province is located in the Northeast Plain and includes the Songnen Plain and the Sanjiang Plain. Other regions with high susceptibilities were distributed in the Daxinganling, Xiaoxinganling, and Mudanjiang mountainous regions.

### 3.3. Vulnerability Analysis of Snow Disaster

#### 3.3.1. The Characteristics of the Vulnerability Zoning Factors of Snow Disasters in Heilongjiang Province

Figure 9 shows the spatial distributions of urban lights, livestock numbers, crop-sown areas, and highway mileages in Heilongjiang. It can be seen that the urban night lights are between 0 and 63, and that the high values of urban night lights are scattered in prefecture-level cities and urban areas, while urban night lights in country-level cities are relatively low. The livestock numbers are from 498 to 665,511, and the areas with high levels are mainly distributed in Longjiang County, Shuangcheng City, and Zhaodong City, in the southwest of Heilongjiang Province. The crop-sown areas are from 214 to 342,188 km$^2$, and

the areas with high levels are mainly distributed in Songnen Plain and Sanjiang Plain. The highway mileage is between 92.88 and 25,455, and the areas with high levels are distributed in Qiqihar city, Harbin city, Suihua city, and Heihe city, in Heilongjiang Province.

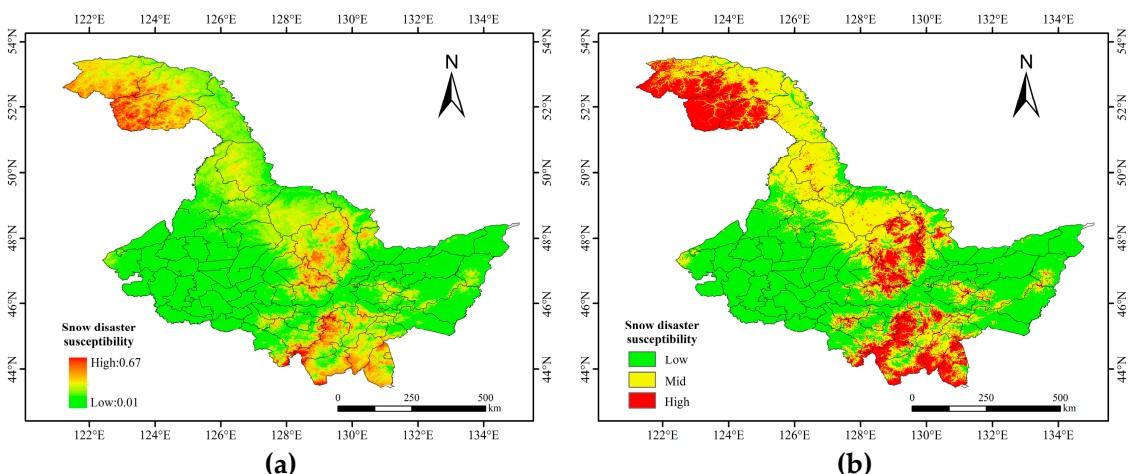

**Figure 8.** Map of snow-disaster susceptibilities in Heilongjiang Province ((**a**): stretch, (**b**): classification).

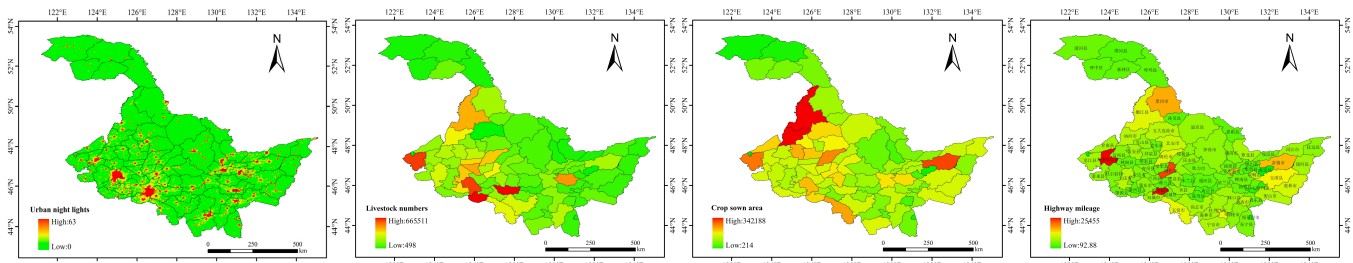

**Figure 9.** Spatial distributions of urban lights, livestock numbers, crop-sown areas, and highway mileages in Heilongjiang.

3.3.2. Results of the Vulnerability Zoning of Snow Disasters in Heilongjiang Province

The weight in Table 2 is used to perform the spatial superposition of the vulnerability factors to obtain the vulnerability zoning results of snow disasters in Heilongjiang Province (Figure 10a). The spatial distribution of the snow-disaster vulnerabilities in Heilongjiang Province is high in the western regions, and low in the central and northwestern regions. The vulnerability index is equally divided into three levels, and the spatial distribution of the vulnerability classification is presented in Figure 10b. High vulnerabilities accounted for 3.0% of the entire Heilongjiang area, and were distributed mainly in the economically developed main urban areas (Qiqihar City, Daqing City, Harbin City, and Suihua City), as well as other small parts of cities and counties; medium vulnerabilities constituted about 27.9% of the entire Heilongjiang area, and were distributed chiefly in Heihe City, Nenjiang County, Nehe City, Longjiang County, Zhaodong City, Wuchang City, Fujin City, Baoqing County, and other cities and counties of Heilongjiang Province; and low vulnerabilities accounted for 69.1% of the entire area of Heilongjiang Province, and were distributed mostly in the northern and central regions of Heilongjiang Province. As shown in the figure, high vulnerabilities were distributed mainly in regions with more urban lights and larger populations, which have been significantly affected by disasters.

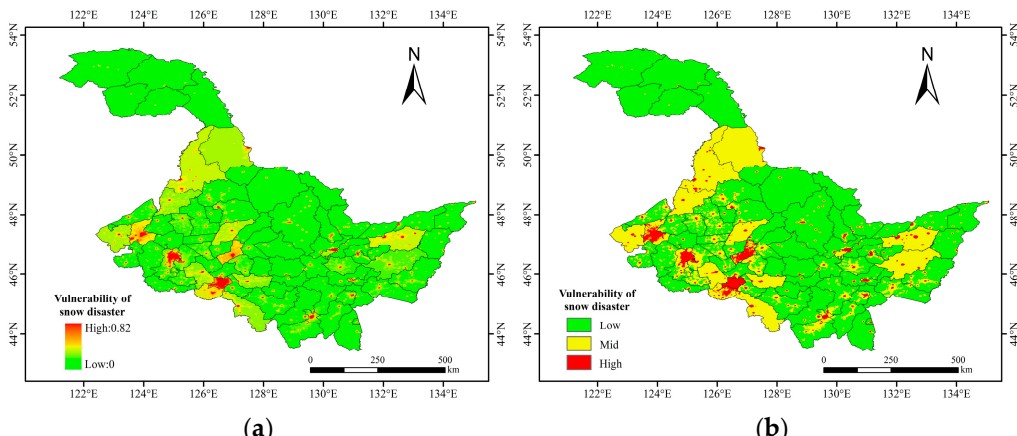

**Figure 10.** Vulnerability zoning map of snow disaster in Heilongjiang Province ((**a**): stretch, (**b**): classification).

### 3.4. Analysis of Snow-Disaster Prevention and Mitigation Capability

3.4.1. The Characteristics of the Prevention and Mitigation Capability Zoning Factors of Snow Disasters in Heilongjiang Province

Socioeconomic factors are the main indicators that affect the disaster prevention and mitigation capability. We selected the economic indicators of the Heilongjiang Province Statistical Yearbook, from 1996 to 2020 that each county has data for, as the factors of the disaster prevention and mitigation capability, and finally selected the total GDP, the per capita disposable income, and the education level. The higher the GDP, the stronger the region's economic strength, and the stronger its ability to fight disasters. The population size of each region is different, and the per capita disposable income further illustrates the level of economic development. Therefore, the per capita disposable income was selected as the disaster prevention and mitigation ability. The level of education represents the level of awareness of the impacts of, and the defenses against, disasters. Therefore, the higher the level of education, the stronger the ability to reduce disasters.

Figure 11 shows the spatial distributions of the total GDPs, per capita disposable incomes, and education levels in Heilongjiang. It can be seen that the total GDP is between 25,538 and 15,324,071, and that the high values of the total GDPs are mainly distributed in Daqing City, Qiqihar City, and Harbin City. The per capita disposable income is between 2514 and 14,850, and the areas with high levels are mainly distributed in Daqing City, Harbin City, Dongning County, and Hailin City. The education level is between 2300 and 73,770, and the areas with high levels are the same as those with high total GDPs.

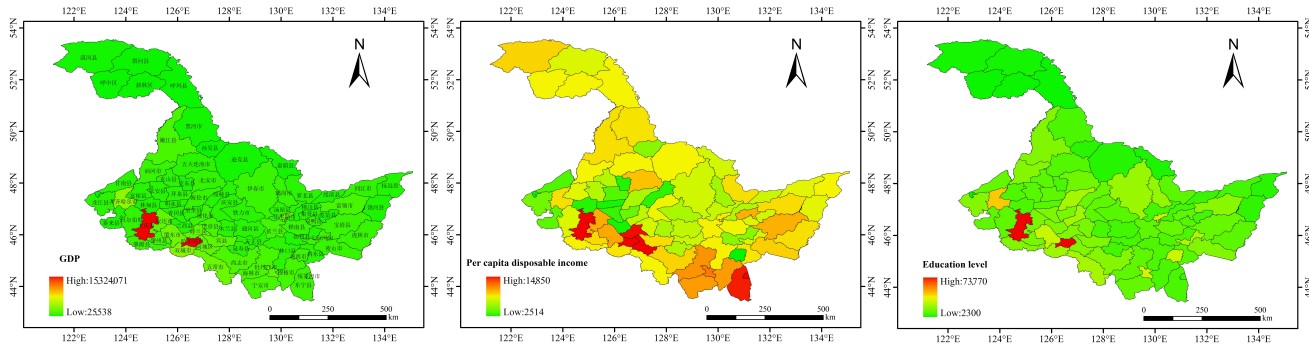

**Figure 11.** Spatial distributions of total GDPs, per capita disposable incomes, and education levels in Heilongjiang Province.

### 3.4.2. Results of the Prevention and Mitigation Capacity Zoning of Snow Disasters in Heilongjiang Province

The weight in Table 2 is used to perform the spatial superposition of the prevention and mitigation capacity factors to obtain the prevention and mitigation capacity zoning results of snow disaster in Heilongjiang Province (Figure 12a). The spatial distribution of the snow-disaster prevention and mitigation capacities in Heilongjiang Province is uneven and is high in the southern regions, and low in the northwestern regions. The prevention and mitigation capacity index is equally divided into three levels, and the spatial distribution of the prevention and mitigation capacity classification is presented in Figure 12b. High disaster prevention and mitigation capabilities accounted for 7.4% of the entire Heilongjiang area, and were distributed mainly in Qiqihar City, Daqing City, Zhaodong City, Hulan District, Harbin City, Acheng District, Shuangcheng City, Mudanjiang City, Jiamusi City, Suifenhe City, and Dongning County; middle disaster prevention and mitigation capabilities constituted 84.0% of the entire Heilongjiang area, and were distributed in most cities and counties in Heilongjiang Province; and low disaster prevention and mitigation capabilities covered 8.6% of the entire Heilongjiang Province area, and were distributed mainly in Sunwu County, Gannan County, Tailai County, Lindian County, Kedong County, Baiquan County, Mingshui County, Qinggang County, Wangkui County, Lanxi County, Suileng County, and Jixi City. The figure shows that the more developed the economy, the stronger the disaster prevention and mitigation capability.

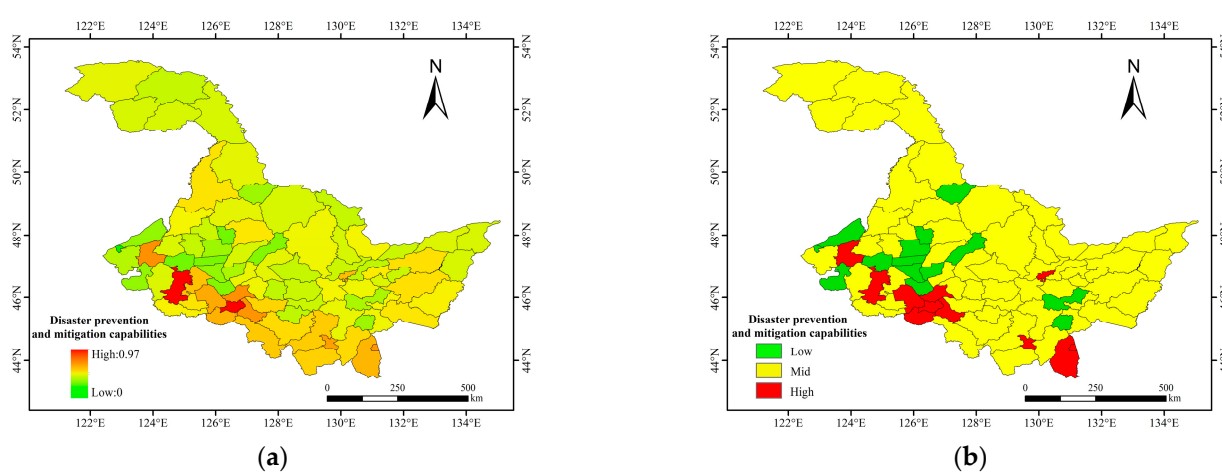

**Figure 12.** Map of Heilongjiang Province's snow-disaster prevention and mitigation capacities ((**a**): stretch, (**b**): classification).

### 3.5. Comprehensive Snow-Disaster Risk Assessment and Zoning

The risk factors, the susceptibility factors, the vulnerability factors, and the prevention and mitigation capacity factors are superimposed, according to the weights, and the comprehensive snow disaster risk index is obtained. The comprehensive risk assessment and zoning results for snow disasters in Heilongjiang Province are shown in Figure 13a. The comprehensive risk index is equally divided into three levels (Table 4), and the spatial distribution of the comprehensive risk classification is presented in Figure 13b.The spatial distribution of the comprehensive snow disaster risk in Heilongjiang Province followed a declining trend from east to west. High risks accounted for 34.3% of the entire Heilongjiang Province area (i.e., an area of 97,000 km$^2$), and were distributed mainly in Hulin City, the eastern part of Mishan City, Shuangyashan City, Qitaihe City, Suifenhe City, Dongning County, Muling City, Ning'an City, Hailin, Mudanjiang City, Yichun City, Jiayin County, Xunke, Hegang, Tangyuan, Jiamusi, Tieli, Mohe County, Huzhong District, Xinlin District, and western Tahe County; medium vulnerabilities covered 45.2% of the Heilongjiang area (i.e., an area of 214,000 km$^2$), largely distributed in eastern Tahe County, Huma County, Heihe City, Nenjiang County, Sunwu County, eastern Wudalianchi, Bei'an, Qing'an, Hailun, Suihua, Mulan, Tonghe, Yilan, Luobei County, Suibin County, Tongjiang City,

Fuyuan County, Fujin, Baoqing, Youyi, Jixian, Linkou, Jixi, Jidong, Shangzhi City, and Wuchang City; and low vulnerabilities accounted for 20.5% of the Heilongjiang Province area (i.e., an area of 162,000 km²), and were distributed mostly in the southwestern region of Heilongjiang Province, including Qiqihar, Daqing, western Suihua, and western Harbin.

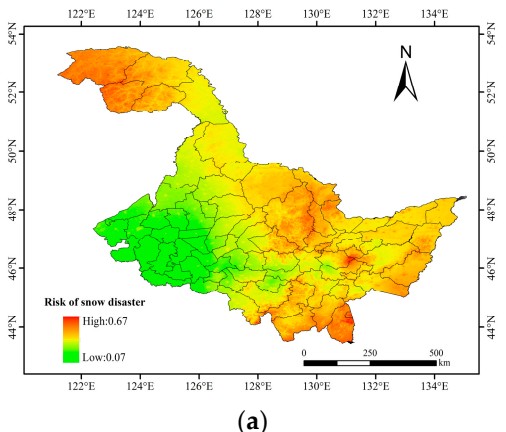
(**a**)

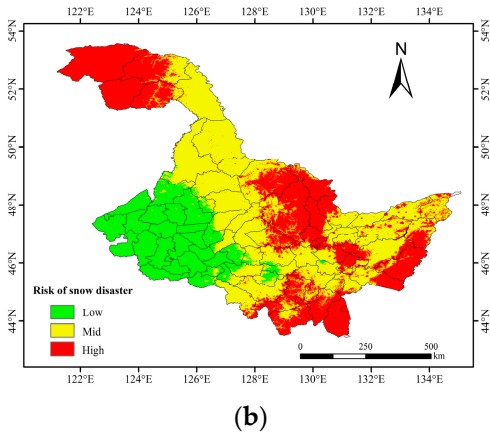
(**b**)

**Figure 13.** Comprehensive risk zoning map of Heilongjiang snow disaster ((**a**): stretch, (**b**): classification).

## 4. Discussion

(1) This article evaluates and zones the snow-disaster risk in Heilongjiang Province from four aspects: risk, susceptibility, vulnerability, and the disaster prevention and mitigation capability. Compared to the snow-disaster risk research at home and abroad, the analysis content of this article is complete. The zoning indicators are relatively complete, which provides a research example for related research and has a certain reference effect. It should be noted that the constituent factors of risk, susceptibility, vulnerability, and the disaster prevention and mitigation capability are not necessarily fixed, and adjustments and replacements can be made without affecting the physical meaning when conducting research in different regions. Generally speaking, six index meteorological indicators, which included the snow depth, the snow-period length, snow days, total blizzard volume, the number of blizzards, and their scores were selected for risk, which are generally easier to obtain; the altitude and land-use types that constitute susceptibility are also easier to obtain. The standard deviations of the slope and terrain are calculated on the basis of altitude data. The NDVI data uses NDVI data; and the factors affecting vulnerability and the disaster reduction capabilities are mainly derived from statistical yearbooks. Since the indicators of the statistical yearbooks in different regions are different, the data in the statistical yearbooks can be replaced when conducting research. For example, the urban lighting index was used in this article because there is no spatial distribution data of populations in the statistical yearbooks. This article uses the MODIS lighting index to replace it.

(2) In the process of zoning, not only can we obtain a spatial distribution map of the different factors that affect not only the risk, susceptibility, vulnerability, and disaster prevention and mitigation capability, but also a zoning map of the risk, susceptibility, vulnerability, disaster prevention and mitigation capability, and the comprehensive disaster risk. Therefore, this method is conducive to the comprehensive analysis of disaster risk. For example, the Sanjiang Plain, in the east of Heilongjiang Province, is a high-risk area for snow disasters, compared with Figure 14, and most of the final comprehensive snow-disaster risk is in a medium-risk area. The main reason is the low susceptibility and the low vulnerability. The comparison and influencing factors can be further analyzed as follows: Although the snow depth, intensity, blizzard volume, and the number of blizzards are relatively large and strong here, it is a plain with a low altitude, small slopes, and small topographical fluctuations. This area is, especially, a primarily agricultural area, basically

consisting of arable land, and there are almost no crops in winter, so the comprehensive risk of snow disaster is low. Therefore, the results of the snow-disaster risk zoning in this paper provide theoretical support for the government and for disaster prevention and relief departments to formulate predisaster mitigation plans, and to help with postdisaster relief decision-making.

(3) Since the selection of zoning factors in this paper will be affected by human factors, on the one hand, and will be limited by statistical yearbook data, on the other, the uncertainty of the factor selection will affect the accuracy of the zoning results. Especially when zoning a large spatial area, it is difficult to obtain a unified socioeconomic index, so it will be restricted. It is recommended to use remote sensing products as much as possible, as with the NDVI and the urban lighting index in this article, which can achieve large-scale spatial risk zoning while ensuring high spatial resolution.

In addition, many meteorological disasters, such as droughts and floods, are related to precipitation data. The precipitation is affected by small-scale weather systems, as well as by the local topography, the underlying surface, and other factors. The spatial interpolation accuracy of precipitation will be affected. The precipitation data in the published precipitation reanalysis data also face accuracy problems. However, there is no good method for the spatial interpolation of precipitation, and research in this area needs to be strengthened.

(4) In the process of zoning in this paper, in addition to the factor selection, there were also problems in calculating the weight of each factor. Because there is no dependent variable, linear regression and other statistical methods cannot be used. The analytic hierarchy process method was used in this article to determine the weights, which are mainly based on the relative importance of each factor and are made by human judgment. Therefore, there are human factors that affect the impact. The results of the zoning will have a certain impact. The analytic hierarchy process is an analytical and discriminative decision-making method, proposed by the American scholar, Saaty, in 1997 [37]. However, although the analytic hierarchy process involves decision-making on data and information, it retains the judgment of the human perception of things and can be completed with less information. The advantage is judging the content of decision-making under the circumstances; however, the disadvantage is that the final result is very dependent on the decisionmaker's judgment of the objective conditions, which has a more obvious subjective intention. When the decisionmaker's perception of things is not accurate enough, it is easy for them to have bias. Therefore, some scholars have proposed methods, such as the multivariate instability index analysis method and the combination weighting method, for weight assignment. We will also conduct further related research.

(5) The current study, based on the meteorological data from 1983 to 2015, reveals the significant reduction in the snow-cover period in Heilongjiang Province over the past 33 years. This reduction is consistent with findings from an analysis of the spatiotemporal variation characteristics of snow in China from 1961 to 2012 [38]. The decrease in the snow-cover period, with interannual variations, has been caused by climate warming over the past 100 years [39]. In addition, the declining trend of the snow period with the latitude variation was similar to that identified in a previous study on the spatiotemporal distribution of snow cover in the northeast region [40]. The snowfall intensity in the east of Heilongjiang Province was higher than in the west, which was consistent with the research results of Huang et al. [30]. The total numbers of blizzards and the total amounts of snowfall in Heilongjiang Province were higher in the east and north, which is similar to the research findings of Zhang [13].

(6) In order to test the results of the snow-disaster risk zoning proposed in this paper, we used the snow-disaster occurrence data in Heilongjiang Province, recorded in the "Chinese Meteorological Disaster Dictionary-Heilongjiang Volume", to compare with this paper. According to the literature, snow disasters in Heilongjiang mostly occur in the eastern parts of the mountains. Traffic snow disasters are more serious, and there are frequent occurrences in Huanan, Boli, Linkou, Yilan, Tonghe, Jixi, Suifenhe, and other

places. The results are similar. At the same time, this article also counts the 2016–2020 snow-disaster loss data recorded by the Heilongjiang Provincial Civil Affairs Department, and the loss, according to the snow disaster, is shown in Figure 14. Compared with the results of the comprehensive risk zoning in this article, it is basically consistent. However, there are some differences in the Lesser Khingan area. The main reason is that the snow-disaster risk in the Lesser Khingan area threatens a high-value area. In the process of zoning in this article, the risk weight is higher, which leads to a higher overall risk. This result also reminds us that, when carrying out disaster risk zoning, we must appropriately increase the vulnerability weight of the disaster-bearing body. In addition, Zhang et al. [13] conducted research on the comprehensive risk of blizzards and found that the high-risk zone is located in the southeast, that the low-risk zone is located in the western region, and that the remaining areas are medium-risk zones, which are also the snow-disaster risks in this paper. These earlier studies provided references for the snow-disaster risk analysis used in this study.

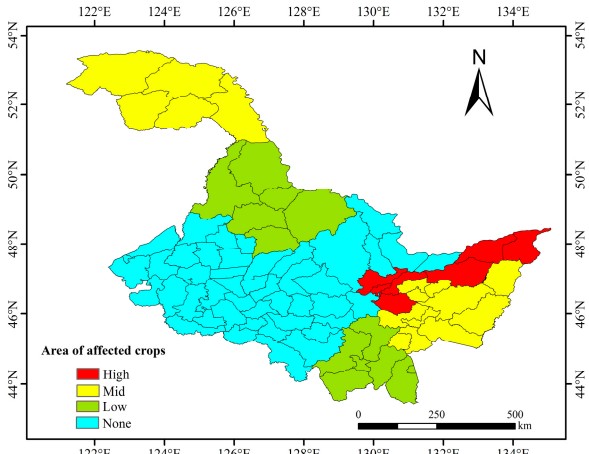

**Figure 14.** Snow-disaster loss data recorded by the Heilongjiang Provincial Civil Affairs Department.

## 5. Conclusions

On the basis of this research, the following conclusions can be made:

1. The snow depth, snowfall intensity, total number of snowstorms, and the total volume of snowstorms in Heilongjiang Province all followed an upward trend with inter-annual changes. The snow depth in the northeast was higher than that in the southwest. The total amount of snowfall, the total number of snowstorms, and the snowfall intensities in the east were all higher than those in the west. The snow-cover period and the number of days with wind speeds >8 m/s decreased with the interannual changes. The snow-cover period in the north was longer than that in the south. The number of days with wind speeds >8 m/s were greater than in the east, but relatively less than in the central area.

2. The snow-disaster risk was higher in eastern and northern Heilongjiang Province. Susceptibilities were higher for the central area and the northern Daxinganling area. Vulnerabilities were higher in the western part than in other areas and were distributed throughout various urban areas. The disaster prevention and mitigation capabilities of economically developed areas were stronger than those in less economically developed areas. The high-risk areas for snow disasters in Heilongjiang Province were located in the northern and eastern regions (e.g., Daxinganling, Yichun, and Mudanjiang), accounting for 34.3% of the total area of Heilongjiang Province. The medium-risk areas of snow disasters were located in the central region (e.g., Heihe, Suihua, Jiamusi, and eastern Harbin), accounting for 45.2% of the total area of Heilongjiang Province. The low-risk areas were located in the southwest of Harbin (e.g., Qiqihar, Daqing, western Suihua, and western Harbin), accounting for 20.5% of the entire Heilongjiang Province area.

**Author Contributions:** Writing—original draft preparation, H.L.; writing—review and editing, L.Z. and S.Z.; formal analysis, W.X. All authors have read and agreed to the published version of the manuscript.

**Funding:** This research was funded by the National Natural Science Foundation of China (No. U20A2082; No. 41771067), and was supported by the Natural Science Foundation of Heilongjiang Province of China (No. ZD2020D002).

**Institutional Review Board Statement:** Not applicable.

**Informed Consent Statement:** Not applicable.

**Data Availability Statement:** The data presented in this study are available on request from the corresponding author.

**Acknowledgments:** We gratefully acknowledge the assistance of Huang, Y.T. and Pan, T. in preparing software and photograph. We also thank Liu, Y.T. and Li, X.Z. for valuable discussion.

**Conflicts of Interest:** The authors declare no conflict of interest.

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
