# Peer review of "Snow-Disaster Risk Zoning and Assessment in Heilongjiang Province"

_sustainability, doi:10.3390/su132414010_

Round 1
Reviewer 1 Report
The manuscript "Snow disaster risk zoning and assessment in Heilongjiang Province" may be considered for publication in Sustainability after Major Revision.
Major comments:
- The presented statistical analyzes are faulty. The authors used a linear regression model for trend analysis. This is unacceptable. It is correct to use the Mann-Kendal test to check if there is a trend in the time series. If it exists, apply the Pettitt test to identify the likely point of change in the time series.
- No statistical analysis is presented. Maps only. I suggest that boxplots be constructed to demonstrate the temporal variability of the evaluated variables (i.e. NDVI).
- I believe that a Pearson correlation analysis between the variables is useful to explain some of the results.
- The Discussion is very poor and does not demonstrate the main scientific advances that this work provided. It is necessary to expand it.
Minor comments:
- Tables 2, 3, 4 and 5 are unnecessary. If you want, present them as supplementary material.
- Figure captions are in Mandarin. Please correct them for English.
Reviewer 2 Report
This is an interesting article, but it is not stated what the result of the solution is intended for and for whom (I suggest adding or highlighting it more). The considered elements and criteria of the hierarchical model (Fig.2) are relevant and quantifiable. The mathematical apparatus used seems to be suitable, the methods are described briefly and clearly. In some parts (lines 208 - 211) it needs to be reworded and supplemented for a better understanding of the idea.
The results of the solution are presented in a descriptive and clearer graphic form. I recommend the authors to expand this section and present the results for individual areas, e.g. in the summary table. Similarly, the expression Snow Disaster Risk Assessment and Zoning - rather in the form of a table and then a graphical expression. Why are formulas 5 and 8 different, what is the difference ?? Please clarify.
The discussion of the results is relatively short, I recommend expanding it and mentioning e.g. uncertainties that had to be considered in the calculation, or who and on what basis decided on the parameters in the decision matrices (Tab 2 -5), etc. The conclusions are clear, but I recommend extending them.
Recommendations:
- Extend the introduction and analysis of the current situation in the field of research focus by more resources (mainly focusing on foreign solutions and approaches, not only China)
- In the introductory sections, define unambiguous GAPs, i. in which the research is original; what will be the benefits of the solution; who, when and for what can use the results.
- Chapter 2.3 - why only linear regression was used, to clarify
- Check - on r. 157 you present 7 risks, in tab. 2 is only 6 items (columns)
- The key is to clarify who and on what basis determined the values ​​in tab. 2 to 5.
- Check and explain why there is a difference between formulas 5 and 8, if they are to be the same - then only formula 5 is enough.
- In the description of the methods, I recommend stating what they will be used for in the research and whether there is any connection between them.
- I recommend broadening the discussion about the possibility of further solving the problem and using the results of research - to highlight what and for whom the results are important.
- To include in the literature and some sources from abroad, the problem is certainly being solved elsewhere in the world.
Formal:
- Eq. 1 - clarify what is "a";
- unify formulas 2 and 3
- check for symbolic and linguistic deficiencies.
Reviewer 3 Report
Variables in Equations (2)(3) should be explained.
Make sure your variable setting is consistent over the manuscript. In Equation (4), Y represents the comprehensive disaster risk index, while I represents the comprehensive disaster risk index in Equation (5). Are they the same? In Equation (5), Wa, Wb, Wc, and Wd are the corresponding weights and are calculated using AHP, why are they not shown in Equation (2)and (3)?
In Equation (4), Xi is the impact disaster index, and λi is the weight value. What is “impact disaster index”, I failed to find this term anywhere else in this manuscript. How to determine λi, please explain.
Are Y and Y representing the same variable?
The row and column headers of Table 2-5 should be changed. The notes under tables are unnecessary.
Equations (6) and (7) are two different standardization methods, so please distinguish the X’ij used in these Equations.
Line 208, “We selected Equations (3) and (4) according to the relationship between the factors in the zoning and the suitability of crop planting.” Do you mean Equations (5) and (6)?
The design of factors lacks discussion. For example, using total GDP, per capita disposable income, and education level as the factors of disaster prevention and mitigation capacity is problematic.
The resolution of figures 3-11 are too low. The legend in each figure is presented with Chinese characters.
The literature review is insufficient. The introduction of Heilongjiang should be moved to the case study part. Instead, a comprehensive review of state-of-the-art climate risk assessment studies should be presented. Without a proper literature review, the contribution of this paper is unclear.
The methodology section is poorly written. The choice of AHP as the weighting method is arbitrary. As a subjective weighting method, AHP quantifies the weights of decision criteria based on individual experts’ experiences. The survey data collection process is unclear.
The greatest concern with this type of research article is the validation of the proposed method. It is the authors’ responsibility to convince readers that the proposed method can accurately assess snow disaster risk and be more advanced than existing methods, at least in some aspects.
The authors consider four aspects (risk, sensitivity, vulnerability, disaster prevention and mitigation capability) in the risk assessment model. Apparently, their definition of risk includes the possibility of risk and the consequence of risk. However, the consequence of risk is absent in the discussion section. The authors state that their finding is consistent with previous studies, but the unique value of this paper needs more work to do.
The amount of references is insufficient. More high quality academic papers should be citied.
Round 2
Reviewer 1 Report
The manuscript has been revised as per my suggestions and may be accepted for publication.
Author Response
Thanks a lot for your hard work on our revised manuscript.
We are very grateful to your valuable comments and thoughtful suggestions.
Reviewer 3 Report
There is no i in the right hand side of Equation (2). So all Yi are the same?
Table 2 is unnecessary; just report the RI value you need.
In Table 5, I believe the index “Prevention and mitiga-” and “Comprehensive Snow” are incomplete.
Who give the Scale bij values? Professors? Experts?
I suggest present Equations (6)-(12) in Mathtype format.
The authors made a dramatic improvement on the manuscript. I am basically happy with the current version. The literature review is in a single long paragraph, which is hard to read. It should be better organized.
Please make sure reference number is right after author name(s). For example, in Line 65, the ref. number [12] should follow “Tachiiri et al.”
Round 3
Reviewer 3 Report
See the attachment.

Round 4
Reviewer 3 Report
- Study on single snow disaster risk. (Line 62)
(2) Study on the impact of snow disaster. (Line 86)
(3) Study on comprehensive risk regionalization of snow disaster. (Line 108)
I believe it is better to separate the review of references fall in the three categories above into three paragraphs.
The introduction section should conclude with a brief paragraph that describes the organization of the rest of the paper.